# The Effect of Transcranial Electrical Stimulation on the Recovery of Sleep Quality after Sleep Deprivation Based on an EEG Analysis

**DOI:** 10.3390/brainsci13060933

**Published:** 2023-06-09

**Authors:** Yuhan Wang, Qiongfang Cao, Changyou Wei, Fan Xu, Peng Zhang, Hanrui Zeng, Yongcong Shao, Xiechuan Weng, Rong Meng

**Affiliations:** 1Department of Public Health, Chengdu Medical College, Chengdu 610500, China; wyh17745093573@126.com (Y.W.); qiongfangcao@163.com (Q.C.); weichangyou@cmc.edu.cn (C.W.); xufan@cmc.edu.cn (F.X.); 15281822717@163.com (P.Z.); 2Department of Clinic Medicine, Chengdu Medical College, Chengdu 610500, China; zengriri@163.com; 3School of Psychology, Beijing Sport University, Beijing 100084, China; budeshao@bsu.edu.cn; 4Department of Neuroscience, Beijing Institute of Basic Medical Sciences, Beijing 100850, China

**Keywords:** facial expression, EEG, sleep deprivation, restorative sleep

## Abstract

Acute sleep deprivation can reduce the cognitive ability and change the emotional state in humans. However, little is known about how brain EEGs and facial expressions change during acute sleep deprivation (SD). Herein, we employed 34 healthy adult male subjects to undergo acute SD for 36 h, during which, their emotional states and brain EEG power were measured. The subjects were divided randomly into electronic stimulation and control groups. We performed TDCS on the left dorsolateral prefrontal cortex for 2 mA and 30 min in the TDCS group. These results indicated that the proportion of disgusted expressions in the electrical stimulation group was significantly less than the controls after 36 h post-acute SD, while the proportion of neutral expressions was increased post-restorative sleep. Furthermore, the electrical stimulation group presented a more significant impact on slow wave power (theta and delta) than the controls. These findings indicated that emotional changes occurred in the subjects after 36 h post-acute SD, while electrical stimulation could effectively regulate the cortical excitability and excitation inhibition balance after acute SD.

## 1. Introduction

Insufficient sleep is a common phenomenon in contemporary society. It increases the risk of diseases such as type II diabetes [1], hypertension [2,3], cardiovascular diseases [2,3], mental disorders [4], respiratory diseases [5], and digestive diseases [6]. In addition, it also affects humans’ cognition and performance, having impacts such as alertness [7], making the brain stay awake (for example, delta, alpha, and beta), and reducing arousal [8]. Several studies have demonstrated that sleep deprivation (SD) may affect the cognitive systems that rely on emotion [4]. Magnetic resonance studies have shown that, after acute SD, the cerebral blood flow in the brain regions related to emotional control and cognition decreases emotion and cognition [9]. Its impact shows a significant difference among individuals and is similar to that of alcohol poisoning [10], rendering it detrimental to health. Thus, it can be harmful to individuals and the public.

SD can cause several deficits in working memory, cognitive interference, processing speed, and executive function [11]. However, lack of sleep is inevitable in some occupations, such as medical services [12], need-based shift workers [13], and soldiers [14]. Thus, it has become a serious social problem.

Interestingly, the study of transcranial magnetic stimulation (TMS) combined with an EEG to evaluate the frontal lobe oscillatory activity before and after antidepressant treatment for patients with depression disclosed that the power of the beta band (13–20 Hz) increased, while the power of the theta (4.5–7.5 Hz) and α (8–12.5 Hz) bands decreased significantly [15] after SD. Moreover, the severity of this depression decreased significantly after treatment and led to a significantly activated beta/gamma band response (21–50 Hz) [16]. These results showed that the source intensity of the activity in the middle delta increased after SD, mainly in the parietal and frontal lobes [17]. In contrast, only a few studies have reported the effects of electrical stimulation on EEG frequency bands and channels after SD.

Transcranial direct current stimulation (TDCS) is a non-invasive technique used to regulate cortical activity and excitability. A 2 mA stimulation to the dorsolateral prefrontal cortex (DLPFC) improved the symptoms of depression and anxiety in patients and exerted a positive impact on their sleep quality [18]. These results indicated that the positive electrode connected above the DLPFC is related to working memory, sustained attention, and emotional regulation. TDCS stimulation can also improve negative emotions [19]. Together, studies have disclosed that a single 30 min mAtDCS (left DLPFC anodic stimulation and right DLPFC cathodic stimulation) increases the level of N-acetyl aspartate in the forehead, which is a metabolite related to neuronal regulation and an indicator of neurons [20]. However, little is known about the excitatory effect of TDCS on sleep.

How to effectively improve sleep quality has become an important issue in the field of neuroscience. At present, non-invasive nerve regulation and central nerve stimulants have become the dominant options for this. In the field of non-invasive neural regulation, it has been proven that repetitive transcranial magnetic stimulation (rTMS) plays an important role in improving sleep quality [21,22]. Then, questions have arisen about whether electrical stimulation can improve sleep quality after sleep deprivation and whether it can promote a better recovery of the brain. At the same time, some studies have shown that TDCS can improve negative emotions and depression and anxiety in patients [23,24]. Therefore, we assume that electrical stimulation can effectively promote a better recovery after acute sleep deprivation and can effectively improve emotional states after acute sleep deprivation.

## 2. Methods

### 2.1. Ethics Statement

This study was approved by the Institutional Review Board of Bei hang University (Beijing, China) before it commenced (approval No. 20180040). The study protocol was conducted according to the Declaration of Helsinki. All the participants provided written informed consent prior to their enrolment.

### 2.2. Participants

A total of 34 healthy male subjects with good, habitual sleep were recruited from medical schools. In total, there were 17 people in the control group (age 22.38 ± 2.16, BMI 19.96 ± 2.40) and 17 people in TDCS group (age 21.85 ± 1.01, BMI 21.12 ± 3.47). The inclusion criteria were as follows: (a) right-handed; (b) good sleep habits; (c) no sleep disorders; (d) regular dietary habits; (e) no consumption of any stimulants, such as alcohol and caffeine; and (f) no history of any psychiatric or neurological disorders. The exclusion criteria were: (a) BDI ≥ 45 and BAI > 13, which were used to exclude participants with major depressive disorders or anxiety; and (b) people with sleep disorders. The subjects participated in three trials of 8 h of adequate sleep, 36 h of complete SD, and different sleep recovery methods. One week before the formal test, no high-intensity physical and mental activities were carried out and regular work and rest were maintained.

### 2.3. Experimental Procedures

The thirty-four subjects were randomly divided into a control group and TDCS group. The experimental procedure for the SD was as follows: the participants were required to register one day in advance and ensure that they had adaptive sleep between 22:00 on the first day and 08:00 on the second day. The sleep deprivation experiment began at 08:00 on the second day and ended at 20:00 on the third day. During this period, the subjects were taken care of by the staff to ensure that they remained awake throughout the experiment. After this sleep deprivation, the first EEG and face data were collected for both groups. After the end of the sleep deprivation on the third day, the BC group began to resume sleep at 22:30 until 08:00 on the fourth day. The TDCS group began to resume sleep at 22:30, after being stimulated by TDCS for 30 min from 22:00 to 22:30, until 08:00 on the fourth day. The entire sleep record was monitored by the camera system to ensure that both groups had at least 8 h of restorative sleep. After 08:00 on the fourth day, the TDCS group was stimulated by TDCS again for 30 min, and then the second round of EEG and face data were collected for both groups.

During the whole experiment, the subjects stayed in the sleep laboratory. The subjects were not allowed to engage in hard activities and were not allowed to eat substances or beverages that contained stimulants or could produce excitability. During the whole process of the sleep deprivation, staff were arranged to continuously supervise all the subjects to ensure that they remained awake during this sleep deprivation. Additionally, it was ensured that the same video content was used for the two face data records after the sleep deprivation and recovery sleep (Figure 1).

### 2.4. Electrical Stimulation

In this experiment, the Soterix medical electrical stimulation device mini-CT (1 × 1) was used. The regulation of the nervous system [25] was carried out by an experienced researcher. To stimulate the left DLPFC, the 10–20 International Electroencephalographic Electrode Placement System was followed. The TDCS stimulation mode was selected at 2 mA for 30 min, the anode was placed at point F3, and the cathode was placed at point FP1.

### 2.5. Data Recording and Preprocessing

The results of the digital EEG recording (NeuroScan, changzhou, Jiangsu, China) were recorded two times for each group. A 64-lead EEG cap was placed on the scalp, using the 10–20 montage systems. The electrode impedance needed to be maintained at <10 kΩ and the sampling frequency was 1000 Hz. The average recording time was 7 min. The MATLAB R2013b software (Math Works, Natick, MA, USA) and toolbox EEGLAB v13.0.0 (Swartz Centre for Computational Neuroscience, La Jolla, CA, USA; http://www.sccn.ucsd.edu/eeglab/ (accessed on 5 June 2023)) were used to analyze the data offline. EEG preprocessing removed the useless electrode electrooculogram and electromyography, with a filtering range of 0.1–100 and concave filtering of 48–52 to remove the power frequency interference. The sampling rate was reduced to 512 Hz, the whole brain was averaged for reference, and the data were divided into 2 s. In addition, an independent components analysis was conducted on each participant’s dataset and the artifact was removed according to its ICA results. The EEG preprocessing was as follows: delta (0–4 Hz), theta (4–8 Hz), alpha1 (8–10 Hz), alpha2 (10–12 Hz), beta1 (12–15 Hz), beta2 (15–18 Hz), beta3 (18–25 Hz), and beta4 (25–30 Hz). After this preprocessing, the data were analyzed using their average power spectrum to ensure that the distribution and transformation of the EEG rhythm could be observed more intuitively.

The facial expression stimulation watch video contained one neutral, happy, and surprised background content; then, the FaceReader software (Noldus, Wageningen, The Netherlands) was used to analyze the happiness, sadness, anger, disgust, fear, surprise, and neutral expressions after the sleep deprivation and recovery sleep [26].

### 2.6. Statistical Analysis

For the behavioral data, the mean value of each emotion of 17 subjects was selected and a t-test was used to compare the difference between the two conditions (SD vs. RS) (*p* < 0.05). Due to the asymmetric data distribution, a nonparametric rank-sum test was conducted to analyze the differences in the seven emotions before and after the two states. For the EEG data, a two-factor analysis of variance (ANOVA) was performed to distinguish the EEG differences between the two groups under two conditions (2 × 2). For the post hoc test, a simple effect analysis was used to assess the interaction effect, while a main effect analysis was performed if no interaction effect was detected. Finally, a false discovery rate (FDR) correction was performed and meaningful channels were corrected.

## 3. Results

### 3.1. Facial Expression Results

All the subjects were requested to watch the same video after the SD and RS periods, while their facial expressions were recorded twice (Figure 2). Significant differences were detected between the two groups in their neutral expressions via a two-factor analysis and significant differences were detected between the two sessions in their disgusting expression via a two-factor analysis. The main effect results showed that the proportion of neutral expressions after the RS in the electric stimulation group was significantly larger than that under the SD state (*p* = 0.005) and the proportion of disgusting expressions after the RS was significantly lower than that of the SD for 36h (*p* = −0.018) (Appendix A).

### 3.2. Power Spectrum Analysis

The two-factor analysis identified a significant difference in the channels between the groups and sessions and some channels were significantly different among the average power of each band within the two groups. Specifically, the delta wave presented differences in the group, session, and interaction effects. The group effect exhibited that the channels with differences were mainly concentrated in FC3, FT8, C3, C4, CP3, and CP6. In the session effect, there was no significant difference, except for in FT7, CZ, C1C2, T7, T8, CP1, CP2, TP7, PZ, P3, and other channels that presented significant differences for varied scenarios. In terms of the interaction effects, the AF4.F1, F2, F3, FC1, FC3, C3, CP3, CP5, and CP6 channels showed marked differences. The delta wave presented a significant difference between the groups in their frontal lobes and central regions. In the session effect, the changes were mainly concentrated in the frontal lobe, occipital lobe, and temporal region. In terms of the interaction effects, the changes were dominantly concentrated in a small part of the frontal and central region.

The theta wave had different channels in the group, session, and interaction effects. In the group effect, the double channels FT8 and O2 presented significant differences. In the session effect, except for CZ, C1, T7, CP1, CP2, PZ, and O2, the other channels presented significant differences in their scenarios. In terms of the interaction effects, the channels with differences mainly existed in Fp2, AF3, AF4, F1, F2, F3, F4, F5, F7, FC3, FT7, C3, CP3, CP4, CP5, CP6, P3, P5, P6, P8, and O2. In the session effect, the changes were mainly concentrated in the temporal region, frontal lobe, and occipital lobe. In terms of the interaction effects, different channels were mainly concentrated in the frontal lobe.

The alpha wave was conducted across different channels in the session effect. Except for CZ, C1, CP1, CP2, and PZ, the other channels presented significant differences in their scenarios. The alpha wave main change areas were concentrated in the frontal lobe, temporal region, and occipital region.

The beta wave performance was assessed across channels in various sessions and the interaction effects. Except for CZ, T8, CP1, CP2, PZ, POZ, PO4, and PO6, the other channels presented significant differences in the session effect. In the interaction effect, the channels with differences mainly existed in FP1, AF3, AF4, FZ, F1, F2, F3, F4, F5, F7, FC3, and C3 (Figure 3, Appendix A). The changes in the power of the channels in the session effect were mainly concentrated in the frontal lobe and the channel with differences in terms of the interaction effects was concentrated in the frontal lobe.

To elucidate the detailed differences between two channels, we conducted a post hoc test on the different channels with main and interaction effects, respectively. The results showed that the average power of the channels with the delta wave, including FC3, FT8, C3, C4, CP3, and CP6, was significantly higher in the electrical stimulation group than in the controls. Compared to the theta wave, the average power of the electrical stimulation group for channel FT8 was significantly higher than that of the controls. Furthermore, the average power of channel O2i was significantly lower than that of the controls. In the analysis of the two scenarios, the bands and channels with differences were significantly greater in SD than the RS (Figure 4, Appendix A).

The results showed that, when the group was electrically stimulated, the average power of the delta wave in the AF4.F1, F2, F3, FC1, FC3, C3, CP3, CP5, and CP6 channels was lower than that of SD; the theta wave showed negative enhancements in the Fp2, AF3, AF4, F1, F2, F3, F4, F5, F7, FC3, FT7, C3, CP3, CP4, CP5, CP6, P3, P5, P6, P8, and O2 channels; and the beta wave showed negative enhancements in the FP1, AF3, AF4, FZ, F1, F2, F3, F4, F5, F7, FC3, and C3 channels (Appendix A).

### 3.3. Correlation Analysis

A correlation analysis was used to discover the correlation between the changes in the various expressions, and the channels with differences in each band validated the correlations of their changing trends. Firstly, we found that the theta and alpha waves were positively correlated with the scared expression changes in the P6 (*r* = 0.53, *p* = 0.031) and F7 (*r* = 0.53, *p* = 0.032) channels, respectively, in the TDCS group. Among the channels positively correlated with scared expressions, the channels related to theta changes were mainly concentrated in the parietal brain regions, while the channels related to alpha changes were mainly concentrated in the arterial temporal regions (Table 1, Appendix A).

Secondly, in the transcranial direct current stimulation, there were two emotions that were negatively correlated with facial expressions: happy and disgusted. The AF4 (*r* = −0.56, *p* = 0.022) channel in the delta band and the FP2 (*r* = −0.66, *p* = 0.005), AF3 (*r* = −0.65, *p* = 0.006), AF4 (*r* = −0.74, *p* = 0.001), AF8 (*r* = −0.52, *p* = 0.033), F1 (*r* = −0.54, *p* = 0.026), and P8 (*r* = −0.63, *p* = 0.009) channels in the theta band. The F1 (*r* = −0.56, *p* = 0.020) channel in the alpha band and the AF3 (*r* = −0.67, *p* = 0.004) and AF4 (*r* = −0.69, *p* = 0.003) channels in the beta band were negatively correlated with changes in happy expressions. In addition, the FC1 (*r* = −0.56, *p* = 0.023), CP5 (*r* = −0.53, *p* = 0.029), and P5 (*r* = −0.58, *p* = 0.016) channels in the delta band showed a negative correlation with disgusted expressions. The CP5 (*r* = −0.56, *p* = 0.022) and P5 (*r* = −0.64, *p* = 0.006) channels in the theta band also showed a negative correlation with disgusted expressions.

At the same time, we found that, in the channels negatively correlated with happy expressions, the changes in the delta waves were mainly concentrated in the frontal pole, and the changes in the theta waves were also mainly concentrated in the frontal pole and frontal lobe. The changes in the alpha waves were mainly concentrated in the frontal lobe, while the changes in the beta waves were mainly concentrated in the frontal pole. For the channels negatively correlated with disgusted expressions, the changes in the delta waves were mainly concentrated in the frontal pole, while the changes in the theta waves were mainly concentrated in the central region (Table 2).

In the control group, there were two bands that were positively correlated with three emotions. Specifically, the delta, C3 (*r* = 0.67, *p* = 0.004), and C5 (*r* = 0.57, *p* = 0.019) channels were positively correlated with changes in sad expressions, while the F3 (*r* = 0.53, *p* = 0.032) channel was positively correlated with changes in scared expressions. In the theta wave, the CP4 (*r* = 0.57, *p* = 0.020) channel in the theta band was positively correlated with changes in surprised expressions. The brain regions with positive correlations between the delta wave and sad expressions were mainly concentrated in the central regions, while the brain regions with positive correlations with scared expressions were mainly concentrated in the frontal regions. Finally, in the theta wave, the regions that showed positive correlations with surprised expressions were concentrated in the central regions (Table 3, Appendix A).

In the control group, negative correlations were exhibited. In the delta band, the AF4 (*r* = −0.54, *p* = 0.028), F1 (*r* = −0.53, *p* = 0.031), and FC1 (*r* = −0.51, *p* = 0.039) channels and happy expression changes were correlated, the CP4 (*r* = −0.51, *p* = 0.039) channel was correlated with sad expression changes, and the C3 (*r* = −0.63, *p* = 0.008) and C5 (*r* = −0.58, *p* = 0.017) channels and superior expression changes presented a negative correlation. In the theta band, the F2 (*r* = −0.52, *p* = 0.036), F4 (*r* = −0.51, *p* = 0.040), and CP4 (*r* = −0.60, *p* = 0.012) channels were correlated with happy expression changes and the CP6 (*r* = −0.50, *p* = 0.043) channel was negatively correlated with expression changes in distraction. The absolute value range of all the correlation coefficients was 0.5–0.74, indicating a medium degree of association. The main concentration area of delta and happy was in the frontal pole, the channels with negative correlations with sad were mainly concentrated in the central regions, and the channels with negative correlations with surprised were mainly concentrated in the central regions. In the theta wave, the change channels related to happy were concentrated in the frontal pole and the change channels related to disgusted were concentrated in the central regions (Table 4).

Combined with a correlation analysis, we found that the channels of the EEG related to happy expression changes were maximal and negatively correlated in the control and electric stimulation groups. The brain regions were mainly concentrated in the frontal lobe and frontal pole. The electric stimulation group consisted of abundant bands related to expression changes, which were mainly concentrated in happy expressions, showing a negative correlation (Table 1, Table 2, Table 3 and Table 4). Importantly, a negative correlation was established between the channels with differences and disgusted expressions in the delta band.

## 4. Discussion

The changes in the EEG and facial expressions and their correlation in the participants after acute SD were studied under different intervention methods. The data show that TDCS can significantly improve EEGs and facial expressions post-acute SD; for example, compared to natural sleep recovery, the proportion of disgusted expressions decreased after the recovery period, while the proportion of natural expressions increased. Previous evidence has suggested that poor sleep quality can interfere with emotional processing [27]. Additionally, repeated transcranial magnetic stimulation (rTMS) is effective for treating generalized anxiety disorder (GAD) and transcranial direct current stimulation (TDCS) is effective for treating social anxiety disorder (SAD) and GAD [28]. Several studies have shown that electrical stimulation can regulate emotions [29] and TDCS may affect the inhibitory response of healthy samples to negative emotional stimuli [30], which is consistent with our research results. Therefore, we speculate that electrical stimulation combined with natural sleep can improve moods after acute SD, which is an effective, non-invasive neural regulation method.

The EEG data analysis demonstrated that, compared to natural sleep recovery, electrical stimulation can significantly weaken the slow wave. Some studies have shown that the application of bilateral TDCS to the sensorimotor cortex has an impact on the brain and its network activities. They found that TDCS can reduce the left frontal alpha, beta, and gamma power, and increase the global connectivity. Alpha, beta, and gamma power and increasing global connectivity were demonstrated here, especially in the delta, alpha, beta, and gamma frequencies [31]. At the same time, studies have found that rTMS enhances the left parietal lobe and left prefrontal region theta with an electroencephalogram and the left parietal and bilateral prefrontal regions alpha interposition phase synchronization with an EEG. These studies have indicated that information storage requires activity in the parietal and prefrontal regions, which are related to behavioral responses [32]. In our research, we found that the power of the delta band increased in the left parietal region, the power of the theta band decreased in the right occipital region, and the power of the right frontotemporal channel increased. These changes could be attributed to acute SD, which affects the cerebral cortical excitability [33]. To some extent, these results are consistent with the findings of the former studies. Cortical excitability refers to the level of excitation or the inhibition of intracellular neuron interactions [34] and circadian rhythms, which affect the regulation of cortical excitability/inhibition balance [35]. On the other hand, the anodic current of TDCS promotes the depolarization of neurons and enhances the excitability of the cerebral cortex [36], while the cathodic current has the opposite effect. Moreover, electrical stimulation can regulate the cortical excitation–inhibition balance [37], affecting circadian rhythms. Taken together, these results showed that electrical stimulation can regulate the cortical excitability and excitation–inhibition balance after acute SD, thus affecting the cortical activation in the left parietal region and causing a power enhancement in the low-frequency band. However, previous studies have proven that the low-frequency brain rhythm can quickly reach the real sleep state [38]; thus, it could be inferred that electrical stimulation combined with natural sleep can improve sleep quality. Similarly, electrical stimulation during REM can help sleep [39].

Finally, we found a negative correlation between the changes in disgusted expressions and slow waves. Studies have validated that transcranial direct current stimulation (TDCS) can enhance facial processing in a causal manner, and the anode TDCS on the human occipital cortex enhances facial and object perception and memory [40]. Therefore, we can infer whether subjects’ feelings and emotions can be affected after acute SD [41], and the parietal lobe mainly supports sensory inputs, multi-sensory and sensorimotor integration, and other functions [42]. In addition, CP5 and P5 are located in the parietal lobe, which explains this correlation. It is worth noting that this is consistent with our research results. After electrical stimulation, the excitability of the parietal cortex was regulated, sensory integration was improved, and negative facial expressions were reduced. In addition, studies have also proven that emotional states can be evaluated based on the EEG frequency band [43], and TMS studies have shown that emotional processing interacts with the inhibitory function of the dorsolateral prefrontal cortex (DLPFC) [44], providing evidence for a correlation between EEGs and facial expressions. This may well explain why electrical stimulation can improve emotions after acute SD.

In this study, TDCS stimulation was studied in combination with facial expressions and the results are helpful for recovery after acute SD. However, this study also has some shortcomings: the sample size was relatively small because the collection of EEG data takes a lot of time, the people who undergo sleep deprivation are tired, meaning the number of people who were willing to participate in the whole process of the experiment was limited, and the subjects chosen were only men (because the physiological conditions of women makes them prone to some risks). Therefore, a larger sample size should be used for further research to ensure that the results of this study are further improved.

## 5. Conclusions

Electrical stimulation is a non-invasive neuroregulatory tool. Combined with natural sleep conditions, this method can improve moods and sleep states after acute sleep deprivation. It can increase the delta wave power and the right occipital lobe area’s theta power. In addition, electrical stimulation can regulate the excitability of the parietal cortex, thereby improving sensory integration and reducing negative facial expressions.

## 6. Limitations

First, due to the physiological cycles of female subjects and the endocrine disruption caused by acute SD, only male subjects were recruited; hence, these results may have a gender bias. Second, no biochemical markers were included.

## Figures and Tables

**Figure 1 brainsci-13-00933-f001:**
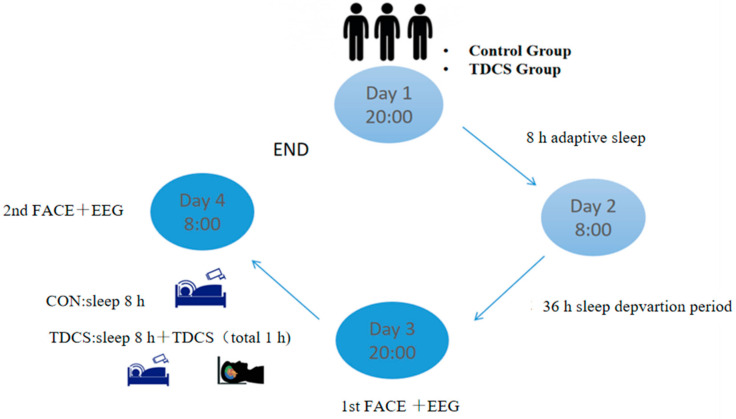
General workflow.

**Figure 2 brainsci-13-00933-f002:**
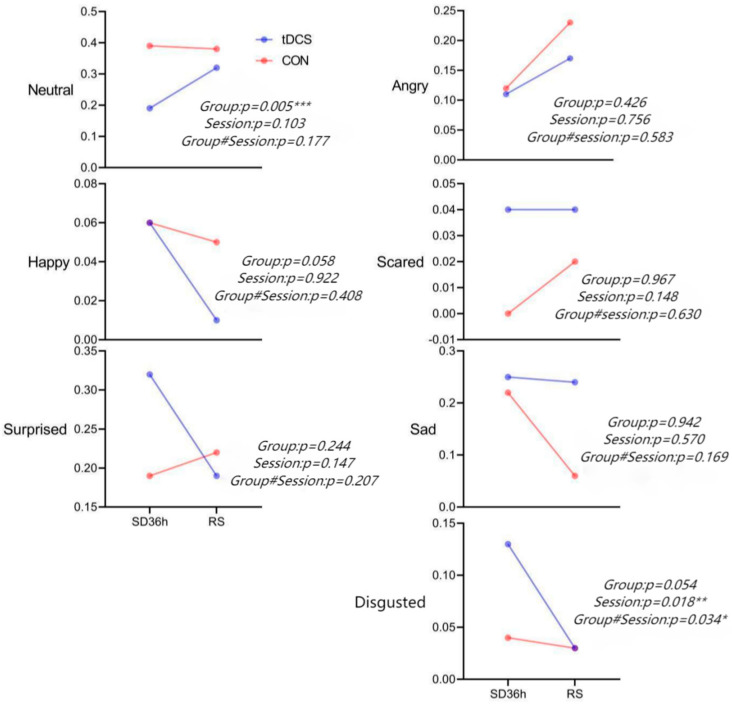
Facial expressions of the two groups after SD36 h and RS (#: Indicates the interaction effect between the group and the session) (*** *p* < 0.01, ** *p* < 0.03, and * *p* < 0.05).

**Figure 3 brainsci-13-00933-f003:**
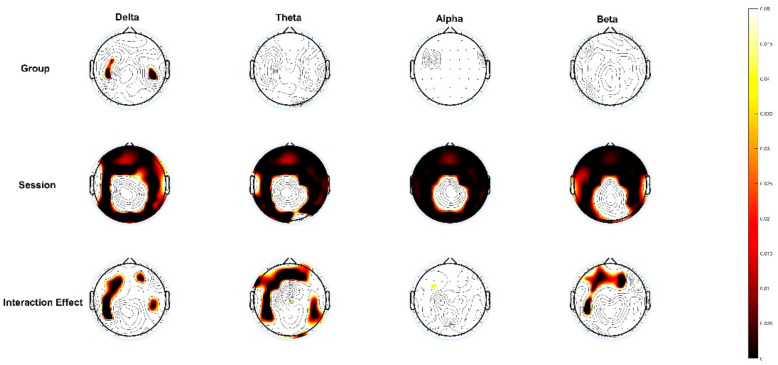
Channel topographic map of differences in the group, session, and interaction effects after two-factor analysis, *p* < 0.05.

**Figure 4 brainsci-13-00933-f004:**
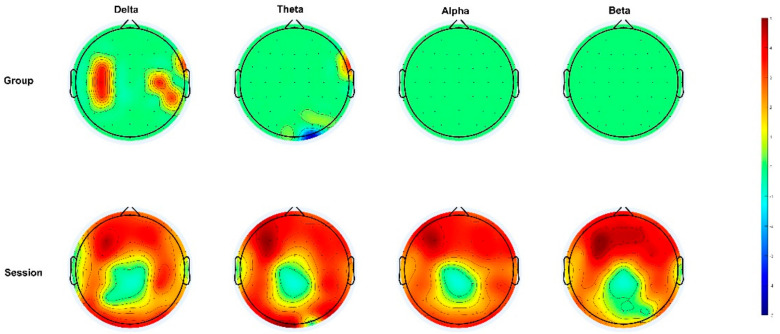
Post hoc comparative topographic maps between groups and within a session.

**Table 1 brainsci-13-00933-t001:** Positive correlation summary table for transcranial direct current stimulation.

Facial Expression	Scared	
	Channels	Encephalic Region
theta	P6	parietal
alpha	F7	anterior temporal

**Table 2 brainsci-13-00933-t002:** Negative correlation summary table for transcranial direct current stimulation.

Facial Expression	Happy		Disgusted	
	Channels	Encephalic Region	Channels	Encephalic Region
Delta	AF4	frontal pole	FC1, CP5, P5	frontal pole
Theta	FP2, AF3, AF4, AF8,F1,P8	frontal pole frontal lobe	CP5, P5	central
Alpha	F1	frontal lobe	--	--
Beta	AF3, AF4	frontal pole	---	--

**Table 3 brainsci-13-00933-t003:** Positive correlation summary table for control group.

Facial Expression	Sad		Surprised		Scared	
	Channels	Encephalic Region	Channels	Encephalic Region	Channels	Encephalic Region
Delta	C3, C5	central	--	--	F3	frontal
Theta	--	--	CP4	central	--	--

**Table 4 brainsci-13-00933-t004:** Negative correlation summary table for control group.

Facial Expression	Happy		Sad		Surprised		Disgusted	
	Channels	Encephalic Region	Channels	Encephalic Region	Channels	Encephalic Region	Channels	Encephalic Region
Delta	AF4, F1, FC1	frontal pole	CP4	central	C3, C5	central	--	--
Theta	F2, F4, CP4	frontal pole	--	--	--	--	CP6	central

## Data Availability

All data will be made available upon reasonable request.

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
