# Peer review of "The Effect of Transcranial Electrical Stimulation on the Recovery of Sleep Quality after Sleep Deprivation Based on an EEG Analysis"

_brainsci, 2023, doi:10.3390/brainsci13060933_

Round 1

Reviewer 1 Report

"Effect of transcranial electrical stimulation on the recovery of sleep quality after sleep deprivation based on EEG analysis" includes an important science content, but the paper needs a major revision:

1. the sample size is very small - may be too small

2. language has to be edited

3. methods, results, discussion & conclusion has to be improved

   (it should be clearer and should be extended)

   especially results - there are a lot of tables, but not enough content and         description

language has to be edited - the sentences, grammar & orthography

(should be edited by a native speaker)

Author Response

Replies to the questions and recommendations of the first reviewer

  1. the sample size is very small - may be too small.

Response: Thank you for your suggestion. On the one hand, because the content of this experiment is sleep deprivation, too many subjects may affect the ethical review. On the other hand, too large sample size is not affordable for our current situation. Finally, we used G*power to calculate and validate the sample size and found that each group of 17 people and two groups were sufficient.

  1. language has to be edited.

Response: Thank you for your suggestion. We are committed to language and readability, and we also involve native English speakers in language correction. See attached proofread certificate.

  1. methods, results, discussion & conclusion has to be improved(it should be clearer and should be extended), especially results - there are a lot of tables, but not enough content and description.

Response: We have revised the corresponding parts in the original text, made trade-offs in the content, retained some parts, and provided more detailed explanations. We revised the relevant content. Also updated the correlation analysis section of the results, from lines 268 to 356, and retained four tables. The remaining table contents removed to supplementary materials. Secondly, we revised the discussion section, see lines 358 to 368 and 410 to 423, and added evidence from lines 388 to 393.

Reviewer 2 Report

The paper reports a study about the effects of tDCS in a group of males after sleep deprivation. The manuscript is interesting and the methods are clearly stated.  I thin the manuscript reports an interesting study with interesting results.

-However I think the authors have too much enthusiasm in the description of the results and they should reduce it. they stated their limits but I think the conclusions are too confident.

-Moreover I think that other limits might be consider like the selection of participants and the evaluation of inclusion/exclusion criteria. Indeed I think they should include more clearly what they evaluated to state that participants were healthy males.

Finally there are some minor points:

- tables captions are missing

- Are there all the tables necessary? I think some tables should be move to supplementary material

- Please change Figure 1 because you also included women in the figure but your sample is composed only by men.

Author Response

Replies to the questions and recommendations of the second reviewer

1.the authors have too much enthusiasm in the description of the results, and they should reduce it. they stated their limits, but I think the conclusions are too confident.

Response: Thank you. We have made changes to the relevant content and have retained some of it.We made changes to the Correlation analysis section of the results, from lines 268 to 356, and retained four tables. The remaining table contents serve as supplementary materials. Secondly, we made changes to the discussed lines 358 to 368 and 410 to 423 and added evidence from lines 388 to 393.

  1. I think that other limits might be consider like the selection of participants and the evaluation of inclusion/exclusion criteria. Indeed I think they should include more clearly what they evaluated to state that participants were healthy males.

Response: Thank you for your suggestion. The inclusion and exclusion criteria for the current experiment have been detailed in the methods section. For further experiments, we will reconsider your suggestion and develop more detailed inclusion and exclusion criteria. For the health issues of the subjects during the entire experimental process, we measured the blood oxygen and heart rate every two hours throughout the entire process and have someone take care of them. At the same time, there will be three fMRI throughout the entire experimental period to ensure their health.

  1. Tables captions are missing.

Response: This may caused by the error of text display, we have double checked. Thanks

  1. Are there all the tables necessary? I think some tables should be move to supplementary material.

Response: Thank you very much for your suggestion. In fact, the main results of this article are Figure 2, Figure 3, and Figure 4. Therefore, the tables are supplementary supporting materials, and we have made changes in the content of the article.

  1. Please change Figure 1 because you also included women in the figure but your sample is composed only by men.

Response: We have made modifications to the content of Figure 1 in the text.

Round 2

Reviewer 1 Report

"Effect of transcranial electrical stimulation on the recovery of 2 sleep quality after sleep deprivation based on EEG analysis" is an interesting paper.

The authors edited it in sense of reviewers.

It can be published after a final check.

"Effect of transcranial electrical stimulation on the recovery of 2 sleep quality after sleep deprivation based on EEG analysis" is an interesting paper.

The language imprved too.

It can be published after a final check.